# Effect of the Silica–Magnetite Nanocomposite Coating Functionalization on the Doxorubicin Sorption/Desorption

**DOI:** 10.3390/pharmaceutics14112271

**Published:** 2022-10-24

**Authors:** Alexander M. Demin, Alexander V. Vakhrushev, Marina S. Valova, Marina A. Korolyova, Mikhail A. Uimin, Artem S. Minin, Varvara A. Pozdina, Iliya V. Byzov, Andrey A. Tumashov, Konstantin A. Chistyakov, Galina L. Levit, Victor P. Krasnov, Valery N. Charushin

**Affiliations:** 1Postovsky Institute of Organic Synthesis, Russian Academy of Sciences (Ural Branch), Ekaterinburg 620108, Russia; 2Mikheev Institute of Metal Physics, Russian Academy of Sciences (Ural Branch), Ekaterinburg 620990, Russia; 3Institute of Immunology and Physiology, Russian Academy of Sciences (Ural Branch), Ekaterinburg 620049, Russia; 4Institute of Natural Sciences and Mathematics, Ural Federal University, Ekaterinburg 620002, Russia; 5Institute of Chemical Engineering, Ural Federal University, Ekaterinburg 620002, Russia

**Keywords:** nanocomposites, Fe_3_O_4_ nanoparticles, SiO_2_, PMIDA, doxorubicin, sorption/desorption, DFT calculations

## Abstract

A series of new composite materials based on Fe_3_O_4_ magnetic nanoparticles coated with SiO_2_ (or aminated SiO_2_) were synthesized. It has been shown that the use of *N*-(phosphonomethyl)iminodiacetic acid (PMIDA) to stabilize nanoparticles before silanization ensures the increased content of a SiO_2_ phase in the Fe_3_O_4_@SiO_2_ nanocomposites (NCs) in comparison with materials obtained under similar conditions, but without PMIDA. It has been demonstrated for the first time that the presence of PMIDA on the surface of NCs increases the level of Dox loading due to specific binding, while surface modification with 3-aminopropylsilane, on the contrary, significantly reduces the sorption capacity of materials. These regularities were in accordance with the results of quantum chemical calculations. It has been shown that the energies of Dox binding to the functional groups of NCs are in good agreement with the experimental data on the Dox sorption on these NCs. The mechanisms of Dox binding to the surface of NCs were proposed: simultaneous coordination of Dox on the PMIDA molecule and silanol groups at the NC surface leads to a synergistic effect in Dox binding. The synthesized NCs exhibited pH-dependent Dox release, as well as dose-dependent cytotoxicity in in vitro experiments. The cytotoxic effects of the studied materials correspond to their calculated IC_50_ values. NCs with a SiO_2_ shell obtained using PMIDA exhibited the highest effect. At the same time, the presence of PMIDA in NCs makes it possible to increase the Dox loading, as well as to reduce its desorption rate, which may be useful in the design of drug delivery vehicles with a prolonged action. We believe that the data obtained can be further used to develop stimuli-responsive materials for targeted cancer chemotherapy.

## 1. Introduction

Cancer is regarded as the leading cause of death in every country in the world. For example, according to the World Health Organization, in 2019, cancer was the first or second leading cause of death in patients under the age of 70 in 112 out of 183 countries [1]. Therefore, at present, great attention is paid to increasing the efficiency and reducing the toxicity of known antitumor agents (doxorubicin (Dox), paclitaxel (Ptx), methotrexate (Mtx), camptothecin (Cpt), etc.), and, in particular, to the design of new systems of their delivery to tumor tissue. The development of micro- and nanocomposite (NC) materials based on mesoporous SiO_2_, which is characterized by a high sorption capacity for antitumor agents and biocompatibility, is one of the main approaches to the design of drug delivery vehicles [2,3]. The efficiency of such materials can be increased by introducing additional components into them: polyethylene glycol or other polymers that impart a “stealth” effect to the particles, which makes it possible to reduce the probability of their absorption by the reticuloendothelial system [4]; molecules providing active drug targeting to the tumor [5,6]; photosensitizers capable of generating singlet oxygen to enable photodynamic therapy [6,7]; photothermal agents for photothermal therapy [7]; low-molecular-weight compounds or polymers (biopolymers) capable of stimuli-responsive release of material in a tumor, and luminophores, which make it possible to visualize the accumulation of particles in the body [7,8,9,10]. One of the most promising directions is the preparation of multimodal composite materials based on SiO_2_ and magnetic nanoparticles (MNPs). The presence of a magnetic core in NCs enables simultaneous visualization of their distribution in the body using magnetic resonance imaging (MRI) [11,12,13,14,15] or magnetic particle imaging (MPI) [16] and tumor therapy by heating tissues above 42 °C with a high-frequency magnetic field (magnetic hyperthermia effect) [17,18], NIR irradiation (photothermal effect) [19,20,21], or high-intensity focused ultrasound (HIFU) [22]. Moreover, the release of cytostatic agents from the delivery vehicle can be magnetically controlled [15,23,24,25,26,27] and additionally pH- [28,29,30,31,32] and glutathione-mediated [33]. As a result, a synergistic effect of hyperthermia and chemotherapy for cancer treatment can be achieved.

Dox is a broad-spectrum anthracycline antibiotic that is one of the most commonly prescribed drugs for cancer treatment. Since it has a rather pronounced cardiotoxicity, obtaining its new dosage forms that can reduce the negative effects of therapy remains an urgent task to this day. In materials based on Fe_3_O_4_ MNPs and SiO_2_, Dox can be fixed covalently (most often when using alkoxysilane derivatives) [10,31,32] and also due to noncovalent interactions with the surface functional groups [4,5,6,7,8,9]. Immobilization of Dox on the surface due to sorption seems to be more preferable, since it is not associated with a change in the structure of the Dox molecule (and, accordingly, a change in its biological activity), which can occur during chemical coupling with activated functional groups on the material surface. In most cases, for materials with a SiO_2_ shell (primarily mesoporous), due to the presence of a developed porous hydrophilic surface rich in active sites, it is possible to achieve a higher level of drug loading [7,8,9,19,34,35,36,37] in comparison with nanomaterials with coatings based on polyethylene glycol (PEG) [8,15,38], polyethyleneimine (PEI) [5,29,39], albumin [40], chitosan [41], polymethyl methacrylate (PMMA) [42], polypyrrole [43], Pluronic F127 [44,45], PEG–PEI copolymers [4], polyamidoamine (PAA) and PEG copolymers [46], other copolymers [29,47,48], and low-molecular-weight compounds [49,50,51,52].

It should be noted that, despite the abundance of works on the synthesis of SiO_2_-based NC materials as drug delivery systems, very little attention has been paid to a comprehensive study of the features of Dox sorption on materials depending on their size, morphology, or the presence of various functional groups on their surface. So, the purpose of this work was to synthesize a number of NC materials based on Fe_3_O_4_ MNPs with SiO_2_ or aminated SiO_2_ coatings of various structures, including those using *N*-phosphonomethyliminodiacetic acid (PMIDA) in the synthesis, to evaluate the efficiency of Dox sorption/desorption depending on the presence of various functional groups on their surface, including the use of quantum chemical calculations in the framework of the electron density functional theory (DFT), and to study NC cytotoxicity in in vitro experiments.

## 2. Materials and Methods

### 2.1. Materials

FeCl_3_ × 6H_2_O and FeSO_4_ × 7H_2_O (Sigma-Aldrich, St. Louis, MO, USA), *N*-(phosphonomethyl)iminodiacetic acid (PMIDA, Sigma-Aldrich, St. Louis, MO, USA), tetraethoxysilane (TEOS, Alfa Aesar, Heysham, Lancashire, UK), (3-aminopropyl)trimethoxysilane (APTMS, Alfa Aesar, Heysham, Lancashire, UK), and doxorubicin hydrochloride (Dox, Sigma-Aldrich, St. Louis, MO, USA) were used.

### 2.2. Synthesis of Starting MNPs ***1*** and Their Stabilization with PMIDA (MNPs-PMIDA ***2***)

The synthesis was carried out by analogy with [53,54], but with minor changes. A solution of FeCl_3_ × 6H_2_O (2.335 g, 8.64 mmol) and FeSO_4_ × 7H_2_O (1.200 g, 4.32 mmol) in water (100 mL) was heated to 40 °C. A saturated NH_4_OH solution (10 mL) was rapidly added under sonification (10 min) and stirring with an overhead stirrer. After 10 min, MNPs were precipitated with a magnet, washed with water (3 × 50 mL) until neutral pH, and dispersed in water (100 mL) to afford a colloidal solution of MNPs **1** at a concentration of 10.8 mg/mL. The subsequent stabilization of MNPs was carried out by analogy with [55,56], but with minor changes. To do this, the resulting colloidal solution of MNPs **1** (1.08 g, 4.66 mmol) in water (100 mL) was heated under sonification to 40 °C. After 10 min, a solution of PMIDA (0.265 g, 1.17 mmol) in water (30 mL) was added. The reaction mixture was stirred with an overhead stirrer at 40 °C for 4 h, and at 20 °C for 16 h. The resulting solution of MNPs-PMIDA **2** was further used for silanization.

### 2.3. Synthesis of Fe_3_O_4_ MNP-Based Composite Materials Coated with SiO_2_ (MNP@SiO_2_ ***3***–***8***) or Aminated SiO_2_ (MNP@SiO_2_-APS ***9***–***14***)

EtOH (80 mL) was added to a solution of MNPs **1** or **2** (0.435 g, 1.88 mmol) in water (40 mL) and heated under sonification to 40 °C, sonicated for 10 min, then a solution of TEOS (0.630, 1.26, and 2.10 mL; 1.5, 3.0, and 5.0 molar excess per MNP, respectively) in EtOH (20 mL) was added, and a saturated NH_4_OH solution (6, 12, and 20 mL, respectively) was added dropwise under sonification (10 min) and continuous stirring. The reaction mixture was stirred at 40 °C for 4 h and at 20 °C for 16 h to afford colloidal solutions of MNP@SiO_2_ **3**–**8** that were further used for amination with APTMS. For this, a solution of APTMS (3 mmol per 1 g MNPs) in EtOH (20 mL) was added to a solution of MNP@SiO_2_ (0.473 mg of MNPs **3**, **6**; 0.544 mg of MNPs **4**, **7**; and 0.651 mg of MNPs **5**, **8**) in water (80 mL) under stirring and sonification at 40 °C for 10 min. The reaction mixture was stirred at 40 °C for 4 h, then at 20 °C for 16 h. MNPs were precipitated with a magnet, washed with EtOH (50 mL) and water (2 × 50 mL), and dispersed in water (40 mL) to afford colloidal solutions of MNPs **9**–**14**.

### 2.4. Synthesis of Dox-Containing Composite Materials (***15***–***26***) and Study of Dox Sorption/Desorption

A solution of Dox (10 mg) in water (1 mL) was added to a solution of NCs **3**–**8** and **9**–**14** (10 mg) in water (9 mL) and mixed using a vortex at 25 °C for 24 h. NCs were centrifuged at 16,000× *g* rpm for 15 min, and the precipitate was washed with water (2 × 5 mL) and dispersed in water (2 mL) to afford NCs **15**–**26**. After Dox sorption, supernatant was analyzed by UV spectrometry (at λ_max_ = 480 nm) and fluorescence spectrometry (at λ_ex_ = 480 nm and λ_em_ = 594 nm). The Dox loading efficiency (LE, wt.%) and loading capacity (LC, wt.%) were calculated by Formulas (1) and (2), respectively, as described in [27]:LE = (*m*_Dox load_ − *m*_Dox_) × 100%/*m*_Dox load_,(1)
LC = (*m*_Dox load_ − *m*_Dox_) × 100%/*m*_nanocomposite_,(2)
where *m*_Dox load_ is the mass (mg) of Dox loaded in the reaction, *m*_Dox_ is the amount (mg) of Dox in supernatant, *m*_nanocomposite_ is the mass (mg) of Dox-containing nanocomposite.

To prepare a phosphate buffer solution (pH = 5.8), Na_2_HPO_4_ × 2H_2_O (1.19 g) and KH_2_PO_4_ (8.25 g) were dissolved in deionized water (1 L). To prepare a phosphate buffer solution (pH = 7.4), Na_2_HPO_4_ × 2H_2_O (2.38 g) and KH_2_PO_4_ (0.19 g) were dissolved in deionized water (1 L). An aliquot (500 µL) was taken from an aqueous colloidal solution of NCs **11**–**14**; NCs were precipitated by centrifugation (10,000× *g* rpm, 10 min), then 3 mL of a buffer solution (pH = 5.8 or 7.4) was added to the precipitate and sonificated. The resulting solutions were stirred at 37 °C for 24 h. To determine the amount of desorbed Dox, the samples were centrifuged (10,000× *g* rpm, 10 min) and analyzed by UV spectrometry and fluorescence spectrometry.

### 2.5. Characterization of Nanocomposites

Transmission electron microscopy (TEM) images were obtained on a Tecnai G2 30 Twin transmission electron microscope (Thermo Fisher Scientific, Waltham, MA, USA). The EDX spectra and Fe, Si and P fractions were determined on an EDX-7000 X-ray fluorescence spectrometer (Shimadzu, Kyoto, Japan) with a range of detected elements: ^11^Na to ^92^U. Microanalyses were performed using a CHN EuroEA 3000 automatic analyzer (EuroVector Instruments & Software, Milan, Italy). The IR spectra were recorded on a Perkin Elmer Spectrum Two FT-IR spectrometer (Perkin Elmer, Waltham, MA, USA) equipped with the ultra-attenuated total reflection (UATR) accessory on the diamond crystal. Quantification of Dox in water solutions was carried out using a UV 2600 spectrometer (Shimadzu, Kyoto, Japan) and a Cary Eclipse fluorescence spectrometer (Agilent, Santa Clara, CA, USA). DLS characterization and zeta potential measurement were carried out on a Malvern Zetasizer Nano ZS instrument (Malvern Instruments, Malvern, UK). The magnetic properties were studied on a vibrating-sample magnetometer in magnetic fields up to 25 kOe at room temperature. The relaxivity of aqueous colloidal solutions of MNPs at a [Fe] concentration ranging from 0.2 mM to 0.2 µM was determined using a relaxometer (ECOTEK Corporation, Houston, TX, USA). The *T*_2_ measurements were carried out using the Carr–Purcell–Meiboom–Gill pulse sequence (90°–echo time [TE]/2–180°–echo) (1 kOe, 4.31 MHz, echo 1 ms).

### 2.6. DFT Calculations of the Total Electronic Energies of the Complexes of Doxorubicin Base with a Small SiO_2_ Cluster and Its Conjugates with APS and PMIDA

All DFT calculations were performed using the ORCA 4.0.1 software (Max Planck Institute for Chemical Energy Conversion, Mülheim a. d. Ruhr, Germany) [57]. The geometric optimization of the initial clusters, Dox, and its complexes was performed using the Becke’s hybrid three-parameter nonlocal exchange functional combined with the Lee–Yang–Parr correlation functional B3LYP [58] and the def2-SVP Alrichs basis set [59] for all atoms. Corrections for dispersion effects were introduced according to the semiempirical scheme of pairwise Grimme correction (D3) [60]. At every optimization step, total electronic energies *E* were calculated at the B3LYP-D3-gCP/def2-SVP level of theory using D3 and geometrical counterpoise (gCP) correction of basis set superposition error [61]. The solvent (water) effects were simulated using a conductor type polarizable continuum model (CPCM) [62].

Before the DFT calculation, we simulated Dox complexes using the VEGA ZZ program [63] by orienting the Dox molecule (in the form of a base) (xyz structure is publicly available in the PubChem databank [https://pubchem.ncbi.nlm.nih.gov/compound/Doxorubicin] (accessed on 19 October 2022) in different positions relative to the SiO_2_ cluster, APS residue, and PMIDA conjugates at a distance of 1.8–2 Å. At first, we optimized the structure of the Dox complexes in the gas phase; then, the found minimum was optimized taking into account water according to CPCM; the general level of the theory is CPCM-H_2_O-B3LYP-D3-gCP/def2-SVP//B3LYP-D3-gCP/def2-SVP. The energy of Dox binding (*E*_b_) with clusters in water was determined by Formula (3) [64]:*E*_b_ = *E*_c_ − (*E*_Dox_ + *E*_cl_),(3)
where *E*_c_, *E*_Dox_ and *E*_cl_ are the total electronic energies of complexes, Dox and clusters, respectively.

For optimized geometries of Dox complexes with the most favorable binding energy *E*_b_, we calculated nonvalent interactions at the pro-molecular level using the method for calculating and imaging noncovalent interactions (NCIPlot) [65].

### 2.7. Assessment of Cytotoxicity of the Obtained NCs

HEK-293 cell culture was obtained from the American Type Culture Collection (ATCC, Manassas, VA, USA). Cells were cultured with DMEM (Gibco, Grand Island, NY, USA) supplemented with 10% fetal bovine serum (BioloT, St.-Petersburg, Russia) and 1% penicillin/streptomycin (BioloT, St.-Petersburg, Russia) in tissue culture dishes in a humidified incubator at 37 °C in an atmosphere of 95% air and 5% CO_2_.

For the MTT assay, 4 × 10^4^ HEK-293 cells in 200 µL of complete medium were plated into the wells of a 96-well plate. After 24 h, the medium was removed from the wells, and 200 µL of a complete medium and 50 µL MTT reagent (DIA-M, Moscow, Russia) at a concentration of 5 mg/mL were added to each well and incubated for 4 h at 37 °C. Formazan crystals were dissolved in DMSO (150 µL), and the absorbance was measured in the wells at 570 nm (Vector-Best, Novosibirsk, Russia).

## 3. Results and Discussion

### 3.1. Synthesis and Characterization of Fe_3_O_4_@SiO_2_ Nanocomposite Materials

A series of NCs based on Fe_3_O_4_ MNPs obtained by co-precipitation and coated with SiO_2_ were synthesized according to Figure 1. 

It is known that various surfactants used to stabilize MNPs significantly affect the morphology and properties of the coating formed on particles during silanization [66].

Before applying the SiO_2_ coating, MNPs are commonly stabilized with organic compounds of various nature such as citric acid [19,67,68], oleic acid [69], poly(vinylpyrrolidone) [19,70], poly(oxyethylene)nonylphenyl ether [71,72], lactose-f-chitosan [73], cetyltrimethylammonium bromide [74,75], and others. Phosphonic acid derivatives can also be used to obtain stable colloidal solutions of MNPs [76,77]. Previously, we have demonstrated that the use of PMIDA allows obtaining materials with a large specific area, which makes it possible to increase the level of covalent or noncovalent immobilization of biomolecules [12,35,53]. Nevertheless, in this work, we have for the first time carried out a comprehensive study of the effect of PMIDA and the amount of TEOS added to the reaction on the formation of the SiO_2_ shell and, accordingly, on the size and morphology of the obtained MNP-based NCs. So, PMIDA was used to stabilize some of the obtained MNPs **1** by analogy with [55,56]. To form a SiO_2_ shell on the initial MNPs **1** and MNPs-PMIDA **2**, various amounts of tetraethyl orthosilicate (TEOS) (1.5, 3.0, 5.0 mol per 1 mol of Fe_3_O_4_) were used to obtain MNP@SiO_2_ **3**–**5** and MNP(PMIDA)@SiO_2_ **6**–**8**. The use of a larger amount of TEOS in the corresponding series led to an increase in the content of the SiO_2_ phase in NCs, which was estimated from the energy dispersive X-ray (EDX) spectral data (Table 1). Some synthesized NC materials were aminated with (3-aminopropyl)trimethoxysilane (APTMS) [54,78] to obtain MNP@SiO_2_-APS **9**–**11** and MNP(PMIDA)@SiO_2_-APS **12**–**14**, respectively.

Based on the data of EDX and especially FTIR spectroscopy, it was shown that during silanization of MNPs with TEOS, the presence of PMIDA promotes an increase in the relative content of SiO_2_ phase in composites **9**–**11** compared with materials **3**–**5** obtained under similar conditions, but without PMIDA (Table 1), and, accordingly, determines their morphology. MNPs obtained without PMIDA did not have a common pronounced shell. Thus, according to TEM data, NCs **3** and **4** were MNPs (10 nm) coated with a thin SiO_2_ shell 2–5 nm thick (Figure 1a) and combined into agglomerates with an average size of ~240 nm according to DLS data (Figure 1b). NCs **5** were chemically cross-linked agglomerates up to 200 nm in size (Figure 1a) (with an average hydrodynamic size of ~270 nm (Figure 1b)).

When starting from MNPs-PMIDA **2**, NCs **6**–**8** were formed with a thick SiO_2_ shell (increasing with an increase in the amount of TEOS). NCs **6**–**8** contained several MNP cores (10 nm) united by a pronounced SiO_2_ shell. The average sizes of NCs **6**–**8** were 26–50 nm (117, 180, and 234 nm, respectively, according to DLS data (Figure 1b)) with a common shell of a thickness of 3.5–5.5 nm (Figure 1a). The synthesized NCs differed in their structure from the core-shell MNPs previously synthesized by our group using PMIDA [12,53]. This can be explained by a new protocol for applying the SiO_2_ coating. Thus, the obtained MNPs **2** were not washed out from excess PMIDA after the stabilization procedure, as in [12,53], and were used in further modifications without additional treatment. The content of PMIDA in MNPs **2** was 0.97 mmol/g, which was ~1.5 times higher than in the samples with a PMIDA monolayer (0.61–0.67 mmol/g) [53,55,56]. In addition, unbound acid was present in the reaction mixture during silanization and probably catalyzed the hydrolysis of TEOS alkoxysilane groups both on the particle surface and throughout the entire volume. As a result, clusters of individual MNPs with an already formed thin SiO_2_ coating were subjected to additional silanization by partially hydrolyzed TEOS from the bulk of the reaction mixture to form a common coating, the thickness of which increased with increasing amount of silane (Figure 1a). It is interesting to note that with an increase in the SiO_2_ proportion in NCs, the degree of their subsequent APS functionalization decreased (Table 1).

It has been previously shown that during silanization, PMIDA molecules are displaced by silane molecules [12,53]. The degree of displacement was determined by the proportion of SiO_2_ in NCs. As a result, after deposition of the SiO_2_ shell, from 65 to 32% of the amount of PMIDA in the initial MNPs remained in the NC materials, and after amination, from 27 to 17% remained (Table 1). The ability of PMIDA to bind to the SiO_2_ surface was demonstrated in [79]. In general, in aqueous solutions, the synthesized NCs obtained using PMIDA had smaller hydrodynamic sizes (Figure 1b), which is more preferable for their medical applications.

The efficiency of nanoparticle silanization was evaluated by FTIR spectroscopy [80], which is often used to study both the features of the molecular structure and to qualitatively and quantitatively assess the composition of nanocomposite materials [81,82,83]. Thus, according to UATR-FTIR spectroscopy, it was possible to semi-quantitatively estimate an increase in the proportion of SiO_2_ in the NC composition depending on the increase in the amount of TEOS or PMIDA [12]. The most characteristic bands for SiO_2_ materials are bands in the region of 1300–750 cm^−1^ (caused mainly by ν_as_ Si–O–Si (~1070 cm^−1^)), ν Si–OH (~950 cm^−1^) and ν_s_ Si–O–Si (~789 cm^−1^)); and for iron oxides, a band in the region of 750–500 cm^−1^ (due to ν Fe–O at ~ 550 cm^−1^). From the ratio of areas under the lines of the absorption bands of Si–O and Fe–O bonds in the IR spectra (*S*_Si–O_/*S*_Fe–O_), one can estimate the relative content of SiO_2_ in NCs by analogy with [82] (Appendix A). Thus, with an increase in the amount of TEOS used for preparation of NCs **3**–**8**, the intensity of the characteristic Si–O bands increased significantly; the *S*_Si–O_/*S*_Fe–O_ values also increased (Table 1, Figure 2).

The UATR-FTIR spectral data correlated with EDX data. It should be noted that with an increase in the proportion of SiO_2_, the number of Si–OH groups increased, as evidenced by an increase in the intensity of the absorption band in the region of 977–949 cm^−1^. A decrease in the amount of PMIDA in NCs **6**–**8** and **12**–**14** could also be observed by a decrease in the intensity of the absorption bands at 1436 and 1402 cm^−1^, which belonged to the symmetric stretching vibrations of the PMIDA COO^−^ group. It is worth noting their shift relative to the bands at 1417 and 1375 cm^−1^ in the spectrum of the initial MNPs **2**, which can be explained by the fact that, in one case, the PMIDA COO^−^ group bound to the surface Fe–OH and in the other case, to Si–OH. The bands related to the asymmetric vibrations of the COO^−^ and PO_3_^2−^ groups of PMIDA [55,56] were superimposed on the bands associated with the vibrations of crystallization water (1615 cm^−1^) and Si–O–Si (1070 cm^−1^) and were difficult to identify.

The obtained materials had high saturation magnetization (*M*_S_), low remanence magnetization (*M*_R_) (up to 0.5 emu/g) and coercive force (*H*_C_) (up to 3 Oe) (Appendix A). The magnetization of NCs decreased in accordance with the content of the nonmagnetic SiO_2_ phase in them.

### 3.2. Synthesis of Dox-Containing Nanocomposite Materials

#### 3.2.1. Study of Dox Sorption/Desorption

Noncovalent immobilization of antitumor agents on nanoparticles can be carried out due to electrostatic and van der Waals interactions, hydrogen bonding, and π-π stacking of the Dox anthracycline fragment with aromatic fragments of the coating material. Therefore, the immobilization efficiency and level are determined by a number of factors, such as particle morphology, surface porosity, charge state or the presence of functional groups on the surface, pH of a solution, and nanoparticle-to-agent ratio.

The Dox molecule (p*K*_a_ = 8.2) is positively charged in aqueous solutions with neutral pH; therefore, it can be adsorbed on the surface due to electrostatic interactions with negatively charged functional groups (for example, silanol ones in the case of materials with a SiO_2_-based surface). Thus, it has been demonstrated that Dox sorption proceeds better at pH = 7.4 compared to lower pH values [19]. The presence of negatively charged groups on the surface of the material promotes the Dox sorption, which has been experimentally demonstrated in a number of works. Nevertheless, an increase in the efficiency and level of Dox loading on hollow aminated (with different degrees of amination) mesoporous SiO_2_ nanoparticles was shown to be increased in comparison with non-aminated ones [34]. Therefore, the question of the effect of the amino groups on the nanomaterial surface on the efficiency of Dox sorption remains open.

It should be noted that in this work, we used NCs **3**–**14** with a negatively charged surface. Due to the presence of an excess amount of silanol groups on the surface, the aminated NCs **9**–**14** also had a negative charge (Figure 1b).

The increased Dox-to-nanoparticle ratio leads to an increase in the level of Dox loading, but to a decrease in its efficiency [9,15,19,47,50]. The option when the maximum loading capacity (LC) is reached at the maximum possible loading efficiency (LE) can be considered as the optimal one [15,47]. In this work, we studied the sorption of Dox on all types of synthesized NCs **3**–**14** (Figure 1). Immobilization was carried out in an aqueous solution with a NCs–Dox mass ratio of 1:1 at room temperature. After Dox sorption on NCs **3**–**14**, the reaction mixtures were centrifuged, and the resulting precipitates were washed with water, dried, and analyzed. The supernatants were analyzed by UV and fluorescence spectrometry.

The Dox content in the powders of NCs **15**–**26** was qualitatively assessed by UATR-FTIR spectroscopy (Figure 3). In the spectra of NCs **18**–**20**, which were obtained based on PMIDA-modified MNPs with a SiO_2_ shell, there were well-defined absorption bands characteristic of Dox: 1720 cm^−1^ (ν C = O of the side chain), 1617 cm^−1^ (ν C = O of anthracene fragment), 1580 cm^−1^ (ν NH), and a group of bands at 1470, 1446, and 1413 cm^−1^ (ν C–H). In the spectra of NCs **15**–**17**, these bands were much less pronounced, which indicated a lower content of Dox in them. Based on the even weaker prominence of these bands in the spectra of NCs **21**–**26** containing APS on the surface, it can be concluded that the Dox loading capacity was much lower than in the corresponding NCs **15**–**17** and **18**–**20**. 

The presence of Dox in the composition of NCs was also confirmed by the appearance of an absorption maximum at 490 nm in the UV-Vis spectra and an emission maximum at 590 nm in the fluorescence spectra (see the Appendix A shows the UV-Vis and fluorescence spectra of NCs **15** and **18** as examples). A significant decrease in fluorescence (fluorescence quenching) is probably associated with the interaction of the Dox molecule with the NC surface.

After sorption, the sizes of NCs were not changed according to TEM data. Figure 4a shows TEM images of NCs **18**–**20** with the highest Dox content as examples.

The main phase of the cores of the described NCs corresponds to magnetite, and it is not changed during modification, which was confirmed by the electron diffraction pattern of NC **18** (JCPDS Card No. (79-0417)) as an example (Appendix A). The mean core size in the synthesized NCs, which was determined from TEM data, was 8.8 nm (Appendix A). The core size histogram was constructed using the Sturges’ method by analogy with [84].

The hydrodynamic diameter of particles with the SiO_2_ shell (NCs **15**–**20**) somewhat decreased, while that of NCs with the aminated SiO_2_ shell (NCs **21**–**26**) actually remained the same (Figure 4b). Aqueous colloidal solutions of NCs obtained using PMIDA had the highest stability. Due to the positive charge of Dox molecules, the negative surface charge of NCs is partially or completely compensated during Dox sorption on them. As a result, the ζ-potential of NCs **15**–**20** becomes weakly negative or even positive (Figure 4b).

Relaxivity (*r*_2_) (Figure 4c) is a key parameter that makes it possible to judge the possibility of using materials as *T*_2_ contrast agents in MRI investigations. We measured the relaxivity *r*_2_ of colloidal solutions of NCs **15**–**20** with a SiO_2_ shell (Figure 5c), which are more stable than their aminated analogues and have a high Dox loading. The studied NCs had the *r*_2_ values comparable to those of the FDA-approved commercial agents: Sinerem^®^ (*D*_h_ ~ 50 nm, *r*_2_ = 60 mmol^−1^ s^−1^), Feridex^®^ (*D*_h_ ~ 200 nm, *r*_2_ = 107 mmol^−1^ s^−1^), and Resovist^®^ (*D*_h_ ~ 60 nm, *r*_2_ = 190 mmol^−1^ s^−1^) [85]. Among the materials studied by us, NC **18** with the smallest diameter had twice the *r*_2_ value, which is comparable with the parameters of previously synthesized MNPs used by us as contrast-enhancing MRI agents [12,13,55]. This indicates the possibility of MRI evaluation of the biodistribution and accumulation of the obtained NCs in in vivo experiments.

#### 3.2.2. Quantitative Estimation of Dox Content

To quantify the content of antitumor agents (in particular, Dox) in the active pharmaceutical substances, liquid chromatography with UV-Vis, fluorescent, mass-spectrometric and electrochemical detection techniques, as well as UV-Vis and fluorescence spectrometry are most often used [86,87]. In the development of drug delivery systems, the Dox content is determined indirectly by UV-Vis spectrometry [28,30,36,47,48,49], fluorescence spectrometry [45,51,52], and high-performance liquid chromatography (HPLC) [15]. The use of the direct method for the quantitative determination of Dox in NCs by the above methods does not seem to be correct, since there is a significant change in the optical properties of Dox when it is in contact with any material. Thus, Dox molecules can interact with surface groups (for example, Si–OH in the SiO_2_-based materials) due to electrostatic interactions with charge transfer, and also form molecular aggregates. As a result, the optical properties (molecular extinction coefficient and fluorescence quantum yield) of adsorbed Dox are changed [88,89,90,91], which will lead to changes in the absorption and fluorescence spectra and distortion of the results of calculations of the amount of Dox on the surface of materials. In this regard, there is a need for the careful development of methods for determination of Dox in filtrates after sorption on any nanomaterials, which, in our opinion, is not given enough attention in the literature. Therefore, a separate task of the work was to study the features of using UV and fluorescence spectrometry for the analysis of Dox in synthesized materials.

To quantify the Dox content in the samples, we used UV and fluorescence spectrometry. Several maxima can be distinguished in the UV spectra of a Dox solution (Figure 5a). For analysis by UV spectrometry, a maximum at 480 nm corresponding to the π→π* transition in the anthracycline Dox fragment was chosen, since there is no absorption of other components of the nanocomposite in this region of the spectrum. In the fluorescence spectra of Dox, there is only one pronounced maximum at 594 nm (Figure 5b) (the emission and absorption spectra with the main maximums at 480 nm coincided in shape). It should be noted that a significant contribution to the analysis error is known to be made by the Dox sorption on the surface of PTFE, glass (or quartz) cuvettes [92], and laboratory glassware, as well as on the material of microfiltration filters and dialysis membranes used to isolate samples (filtrates) from the reaction mixtures. Thus, we failed to clean quartz cuvettes from Dox after analysis, despite repeated washings with water. Therefore, sorption was performed in polypropylene test tubes, and the analysis of aqueous Dox solutions was carried out in acrylic cuvettes (fluorescence spectrometry showed no traces of Dox on the walls of the cuvettes after removal of the analyzed solutions and a single washing with water).

When quantifying the Dox content by UV spectrometry, at first we experimentally determined the molecular extinction coefficient (ε) of Dox. To do this, the calibration curves of the optical density vs. the Dox concentration were plotted in the range of absorbance (*A*) of 0.2–1.0 a.u. (see Appendix A). At a concentration below 0.011 mM, the shoulder at 480 nm increases and the contribution of the band at 493 nm decreases (Appendix A). In most cases, researchers rely on the values ε = 10,067–11,500 M^−1^ cm^−1^ [93,94,95].

However, it is known that Dox in aqueous solutions exists in the monomolecular state and as molecular dimers, which are characterized by different values of ε and luminescence/fluorescence lifetime (1 ns, about 0.2 ns, and about 0.4 fs, respectively) [91]. It has been shown that if in highly diluted solutions Dox is present predominantly in the monomeric form, then already at a concentration of 38 µM, its content is 35%. Therefore, the calculation of the Dox concentration in the studied solutions from the average value ε = 10,850 M^−1^ cm^−1^ can lead to a significant error, especially in the case of a low Dox concentration. Thus, a nonlinear nature of the change in ε was observed (Figure 5c). Depending on the Dox concentration, we observed a change in the shape of the UV-Vis absorption spectrum (Appendix A), which can also be explained by the redistribution of the contributions of the monomolecular form or dimers. In this work, to calculate the Dox concentration, we used the arithmetic mean values of ε for three concentration ranges: ε = 12,900, ε = 12,000, and ε = 11,500 M^−1^ cm^−1^ for solutions with *A* < 0.1, 0.5 > *A* > 0.1, and *A* > 0.5 a.u., respectively.

When analyzing samples with a Dox concentration of less than 0.01 mM (*A* < 0.06 a.u.), the error increases significantly, so the determination of the Dox concentration in solutions by UV spectrometry may be incorrect. For example, when studying the process of Dox desorption from synthesized NCs, it becomes relevant to use fluorescence spectrometry to determine the Dox concentration, since that makes it possible to work with solutions containing 0.001–0.01 mM Dox.

To determine the Dox concentration in solutions by fluorescence spectrometry, a calibration plot of fluorescence intensity at 594 nm vs. Dox concentration (for studied solutions, *A* < 0.1 a.u.) was built (Appendix A). As we have shown, the dependence of the luminescence intensity (as well as that of UV absorbance *A*) on the Dox concentration is not linear. The change in the optical properties is associated with different ratios of the Dox monomolecular form and molecular dimers of two types in solutions. The formation of dimers occurs due to the stacking of Dox aglycone fragments in parallel or antiparallel orientation, which are characterized by different fluorescence lifetimes: 1 ns, about 0.2 ns, and about 0.4 fs, respectively. The ratio of monomolecular and dimer forms is determined by the Dox concentration [87,91]. It should be noted that registration of fluorescence with a lifetime of less than 0.1 ns requires special instrumental capabilities. Thus, the flash xenon lamp, which most devices are equipped with, does not allow recording such signals; therefore, it can be expected that the total fluorescence intensity of the analyzed Dox solutions will be underestimated, which, in turn, will introduce an error in the concentration estimate.

#### 3.2.3. Quantitative Assessment of the Dox Content in NCs after Sorption/Desorption

*Sorption.* Dox loading efficiency (LE, wt.%) and loading capacity (LC, wt.%) (Figure 6a) were calculated based on the UV spectrometry data using Formulas (1) and (2), respectively. It was demonstrated that the initial MNPs **1** and **2** adsorbed Dox within 24 h with LE 7% and 23%, respectively (Appendix A). The SiO_2_-coated NCs adsorbed Dox much more efficiently than their APS-modified analogues, with LC differing by a factor of 1.5–3.0. It should be noted that the Dox loading efficiency for materials **18**–**20** and **24**–**26** obtained using PMIDA was higher by a factor of 1.4–6.0 compared to NCs **15**–**17** and **21**–**23**. Increasing the thickness of the SiO_2_ shell leads to an increase in Dox LC (Figure 6a). The obtained data are consistent with the IR spectroscopy data (Figure 3). The LC value (28%) for NCs **19**, **20** exceeded the loading capacity of some materials described in the literature, including those based on silica-coated MNPs [26,28,35,36] or mesoporous SiO_2_ particles [8,9,34] (LC up to 26%).

*Desorption*. We studied the Dox desorption from NCs **15**–**26**. For this, colloidal solutions of NCs **15**–**26** in phosphate buffer solutions (pH = 5.8 and 7.4) were kept at 37 °C for 24 h. After that, the reaction mixtures were centrifuged, and supernatants were analyzed by fluorescence spectrometry. The pH-dependent desorption of Dox from the synthesized materials was demonstrated (Figure 6b), which can be explained by the fact that the protonation of silanol groups and the hydrophobization of the SiO_2_ surface occur at low pH. In this case, the Dox amino groups are protonated (p*K*a = 8.2), and noncovalent bonds with the surface atoms of NCs are broken. It is interesting to note that for NCs **15**–**17** and **21**–**23** synthesized without PMIDA, the amount of desorbed Dox (as a percentage of the Dox amount loaded onto NCs) was greater than for PMIDA-modified analogues **18**–**20** and **24**–**26**. This can be explained by the fact that in NCs **18**–**20** and **24**–**26**, Dox is retained due to stronger bonding with PMIDA molecules on the particle surface. On the contrary, the presence of APS molecules in NCs promotes the desorption process (for example, for NCs **22** and **23**). It is interesting that in the case of NCs **24**–**26** containing both PMIDA and APS molecules, desorption is lowest, and the difference in the Dox desorption at pH 5.8 and 7.4 levels out. It can be assumed that the binding of Dox in such NCs is largely due to the strong interaction with the phosphonic and/or carboxyl groups of PMIDA. This is consistent with DFT calculations of the binding of the Dox molecule into complexes with the surface functional groups of the synthesized NCs (see Section 3.2.4).

Thus, it has been shown that the presence of PMIDA in NCs makes it possible to increase the Dox loading, as well as to reduce its desorption rate, which may be useful in the design of drug delivery vehicles with a prolonged action. So, slow release can determine the reduced toxicity of the agent when administered intravenously. The drug can begin to act after accumulation in target tissues due to destruction in tumor cell lysosomes, or drug release can be activated by exposure to an alternating magnetic field (by analogy with [27]).

#### 3.2.4. DFT Calculations of the Total Electronic Energies of Dox Complexes with a Small SiO_2_ Cluster and Its Conjugates with APS and PMIDA

To assess the possibility of specific binding of the Dox-base molecule to the surface groups of synthesized NCs, we performed the density functional theory (DFT) simulation of its complexes with small SiO_2_ clusters (which were described in [96] as imitating SiO_2_ nanoparticles), including their conjugates with one (SiO_2_-APS) or two APS molecules (SiO_2_-2APS), as well as with the adsorbed PMIDA molecule (SiO_2_-PMIDA) (Figure 7; see Appendix A). In our study, a small SiO_2_ cluster imitated the outer layer of synthesized nanocomposites. The binding energies (*E*_b_) of Dox with these clusters were calculated at the B3LYP-D3-gCP/def2-SVP level of theory in gas phase and in water using ORCA 4.0.1 software [57] (Appendix A).

Calculations have shown that the *E*_b_ value (–224.88 kJ/mol) for the Dox molecule in the SiO_2_-PMIDA…Dox salt complex is less than that of the most energy favorable SiO_2_…Dox complex (Figure 7d).

This may indicate the strongest binding of Dox to the surface of NCs containing SiO_2_ and PMIDA fragments. Dox coordination in the SiO_2_-PMIDA…Dox complex (Figure 7d) occurs due to the formation of various bonds: an ionic bond between the Dox amino group and the PMIDA phosphonic group; two hydrogen bonds between C(8)–O(H) and C(8)–COCH_2_–O(H) of the Dox molecule and the PMIDA COOH groups; and a hydrogen bond between OH in the Dox pyranosyl group and Si-OH (H-bonded oxygen and hydrogen atoms are highlighted in white). At the same time, the energy of Dox binding into the complex in which PMIDA (see Appendix A, PMIDA-Dox salt A) is higher than *E*_b_ calculated for the SiO_2_-PMIDA…Dox salt (see Appendix A, SiO_2_-PMIDA-Dox salt 2). Thus, it can be said that the simultaneous coordination of Dox with the PMIDA molecule and the silanol groups results in a synergistic effect of decreasing the binding energy of Dox with the SiO_2_-PMIDA cluster in comparison with the binding energies of Dox with the SiO_2_ cluster. Such synergy arises from the fact that Dox is attached to the conjugates not only through ionic and H-bonds, but also through the dispersion and dipole–dipole interactions with PMIDA functional groups, which is clearly seen in the NCI D3 plots (Figure 8, right).

Thus, the *E*_b_ value for SiO_2_-APS…Dox complexes (regardless of the number of APS molecules in them) is higher than for SiO_2_…Dox complexes, and even more so for SiO_2_-PMIDA…Dox. This can explain the fact that the Dox molecule is adsorbed to a lesser extent on NCs containing APS fragments than on NCs with a SiO_2_ or SiO_2_-PMIDA shell.

The calculated data on the energy of Dox binding (Figure 7 and Figure 8) to the functional groups of clusters that mimic the surface of the synthesized NCs are in good agreement with the experimental data on the Dox sorption on these NCs (Figure 6a).

Thus, the lowest sorption was observed in the case of APS-modified NCs **21**–**23** obtained without PMIDA (LC 5–10%), which corresponds to the highest *E*_b_ value of Dox complexes with SiO_2_–APS cluster, *E*_b_ = –73.09 kJ/mol for SiO_2_-APS…Dox complex. Mean values of sorption (LC 20.5%) were achieved for NCs **15**–**17**, *E*_b_ = –186.41 kJ/mol for the SiO_2_…Dox complex, while the highest sorption (LC up to 28%) was achieved for NCs **18**–**20** with the lowest value of *E*_b_ = –224.88 kJ/mol for the SiO_2_–PMIDA…Dox complex, which is due to the combination of the most energy favorable binding of Dox to the phosphonic or carboxyl groups of PMIDA.

### 3.3. Cytotoxicity Assessment

We carried out a comparative study of the cytotoxicity of synthesized NCs **15**–**26** and initial NCs **3**–**14** towards HEK-293 cells using the MTT assay. Appendix A shows cell survival data as a function of Dox concentration (0.1–0.005 mg/mL) calculated from its loading capacity (LC) in the studied NCs. It has been shown that, although according to Figure 7b, PMIDA-containing NCs **18**–**20** exhibited the least desorption of Dox within 24 h, their cytotoxic effect was comparable to the initial Dox at a concentration of 0.1 mg/mL and turned out to be the highest among synthesized NCs. Their PMIDA-free analogues (NCs **15**–**17**) or APS-modified NCs **24**–**26** exhibited markedly lower cytotoxicity at Dox concentrations < 0.1 mg/mL. The aminated NCs **21**–**23**, which did not contain PMIDA, as well as the starting NCs **3**–**14** (see Appendix A), did not exhibit significant toxicity. NCs actively interacted with cells (penetrated into them or were adsorbed on the surface), which is demonstrated in Appendix A.

As a result, it has been shown that the cytotoxic effect of NCs corresponds to their calculated IC_50_ values in the framework of the Hill model by analogy with [97]. The IC_50_ values of NCs **18**–**20** and free Dox are comparable (Table 2, Appendix A). Based on the data obtained, it can be concluded that the cytotoxicity of the studied NCs is determined mainly by the total Dox content in them (Figure 6a), and not by the magnitude of desorption in a model experiment conducted for 24 h at different pH (Figure 6b).

Using confocal microscopy, we demonstrated the possibility of Dox desorption and its accumulation inside cells by the example of cell incubation with NC **18** (Figure 9). 

For better cell visualization, cell nuclei were stained with Hoechst 33,342 (blue) (Figure 9a). NC **18** under confocal microscopy had a weak fluorescence (Figure 9b), which is consistent with the data of fluorescence spectrometry (Appendix A). However, after the cells were incubated with NC **18** for 24 h, we observed additional staining of the cells due to the accumulation of Dox (red) desorbed from the nanoparticles. To obtain an image with maximum detail, the channel mode of a confocal microscope was used, with excitation at 405 (for nuclei stained by Hoechst dye) and 561 nm (for Dox) (Figure 9c,d).

To obtain an informative fluorescent image in ZEN software (2009 (Version 6.0), Carl Zeiss, Oberkochen, Germany), a special lambda mode was used, which allowed determining the emission range with the maximum contrast for this material (Figure 9e). The fluorescence spectra of the investigated nanomaterials in the cells were extracted from the images obtained in the lambda mode (Figure 9f). They corresponded to spectra of Dox at λ_ex_ = 561 nm (Figure 5b).

## 4. Conclusions

In summary, a series of new composite materials based on Fe_3_O_4_ magnetic nanoparticles (MNPs) coated with SiO_2_ or aminated SiO_2_ were synthesized, including the materials obtained using *N*-(phosphonomethyl)iminodiacetic acid (PMIDA). For the first time, the role of PMIDA in the formation of the SiO_2_ coating and in the process of the doxorubicin sorption/desorption was revealed. It has been shown that the use of PMIDA to stabilize nanoparticles before silanization promotes an increase in the relative content of the SiO_2_ phase in Fe_3_O_4_@SiO_2_ nanocomposites compared to materials obtained under similar conditions, but without PMIDA. The conditions of doxorubicin (Dox) immobilization on the synthesized materials were studied by UV and fluorescence spectrometry. It has been shown that the first of these methods is most suitable for studying the process of Dox sorption; the second, for the study of Dox desorption. It has been demonstrated for the first time that the presence of PMIDA on the surface of nanocomposites increases the level of Dox loading (up to 28 mg per 1 mg of MNPs) due to specific binding, while modification of their surface with 3-aminopropylsilane (APS), on the contrary, significantly reduces the sorption capacity of materials (up to 5 mg per 1 mg of MNPs). These regularities were in accordance with the results of quantum chemical calculations. It has been shown that the energies of Dox binding to the functional groups of NCs are in good agreement with the experimental data on the sorption of Dox on these NCs. The mechanisms of Dox binding to the surface of NCs were proposed: simultaneous coordination of Dox on the PMIDA molecule and silanol groups at the NC surface leads to a synergistic effect in Dox binding. Simultaneous coordination of Dox with the PMIDA molecule and silanol groups of the nanocomposite surface leads to a synergistic effect, and the 3-aminopropyl fragment of the APS-modified particles is almost not involved in Dox binding. Synthesized nanocomposites allow achieving a high level of Dox loading and a long pH-determined release. At the same time, the presence of PMIDA in NCs makes it possible to reduce the Dox desorption rate, which may be useful in the design of drug delivery vehicles with a prolonged action. The materials showed dose-dependent cytotoxicity in in vitro experiments on HEK-293 cells. The cytotoxic effects of NCs correspond to their calculated IC_50_ values in the framework of the Hill model. Nanocomposites with a SiO_2_ shell obtained using PMIDA had the greatest effect (their IC_50_ values corresponded to IC_50_ of the starting Dox). We believe that the data obtained can be further used to develop stimuli-responsive materials for targeted cancer chemotherapy.

## Data Availability

Not applicable.

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
