# Peer review of "Effect of the Silica–Magnetite Nanocomposite Coating Functionalization on the Doxorubicin Sorption/Desorption"

_pharmaceutics, 2022, doi:10.3390/pharmaceutics14112271_

Round 1

Author Response

Comments and Suggestions for Authors:

The manuscript of Demin et al. presents a study on the synthesis and properties of magnetic composites functionalized with PMIDA for Doxorubicin (DOX) drug deliver that is employed for target cancer therapy. The development of efficient and biocompatible deliver agents is a hot topic in nanotechnology and biomedicine. This work corresponds to the group of studies in which magnetic nanostructures are considered as efficient drug delivery and as imaging contrast agents (theranostic). The study is focused on the synthesis of Fe oxide in SiO2 nanocomposites and its functionalization for bonding DOX. The work present extended structural, morphological, IR, optical and magnetic characterizations. Also, the study of the T2 relaxations performances as NMR contrast agent, the absorption and desorption and the toxicity assessments are investigated. DFT calculations to understand the bonding process of SiO2 with APS and PMIDA are performed. Hence, it is complete study. However, there are many works that have investigated this same approach with similar results. Even several articles (Coll Surface B 185 (2020) 111274, Biosensors and bioelectronics 92 (2017) 485, RSC Adv 5 (2015) 39179 and early ones) are missing. The novelty of the work in comparison with other works regards the use of PMIDA as agent capping and the DFT calculations. Hence, it is a complete study but novelty is not high.

The manuscript is clear and well written. Some points should be considered that are presented below. The manuscript can be published under minor revisions.

  • The state of art should be completed and a better comparison with other works should be made.

Reply: Thank you very much for your appreciation of our work. In this paper, we tried to emphasize the novelty of our work and have made some additions and revisions in the main text of our manuscripts (see lines 86-89 and 226-229), as well as Abstract and Conclusions; we also added the references you suggested. This manuscript contains a survey of modern papers describing approaches for applying a SiO2 coating on MNPs [refs. 19, 66-74] or concerning the preparation of materials with SiO2 surface and Dox sorption on them [refs. 4-10, 31,32,34-36]. Ref. [Coll Surface B 185 (2020) 111274] that you proposed to cite, was already cited by us in the first version of the manuscript [ref. 19]. We added ref. 37 [Biosensors and Bioelectronics 92 (2017) 489–495] in the revised version. It should be noted that, unlike all the cited articles, we have for the first time carried out a comprehensive study of the effect of PMIDA and the amount of TEOS added on the formation of the SiO2 shell and, accordingly, on the size and morphology of the resulting MNP-based NCs. The role of PMIDA in the formation of the SiO2 coating and in the process of the doxorubicin sorption / desorption was revealed. Based on DFT simulation, the mechanisms of Dox binding to the nanocomposite surface were proposed.

  • Magnetic characterizations. The authors indicate that the nanoparticles are superparamagnetic. In that case, no coercive field neither remanence should be present. However, the authors report non-negligible values of Hc and Mr in line 304 and in the inset of Figure 3, the hysteresis loops are open and they change from sample to sample. Clarify these experiments.

Reply: We do not state that the materials obtained by us are superparamagnetic. The manuscript states that “The low values of Hc suggest that the synthesized NCs are in a state close to superparamagnetic”. To avoid ambiguity, the discussion of superparamagnetism has been removed. Figure 3 has been moved to the Supplementary Materials file (Figure S2).

  • Which is the magnetic field employed in the relaxivity experiments?

Reply: The value of the magnetic field of the instrument was 1 kOe (this information has been added to subsection 2.5. Characterization of Nanocomposites).

  • Show the TEM images and analysis of the Fe oxide nanoparticles.

Reply: To prove that the main phase of cores of the described NCs is magnetite and it is not changed during modification, we provided an electron diffraction pattern of the Dox-loaded nanocomposite (NC 18) (see Supplementary Materials, Figure S3a) as an example; we also provided a histogram of the size distribution for magnetic cores of this NC, which was obtained from the TEM data (see Supplementary Materials, Figure S3b).

  • UATR-FTIR experiments (page 8, lines from 291). Clarify how the quantitative calculations were performed.

Reply: Supplementary Materials (Figure S1) describe the method for qualitative assessment of the ratio of inorganic components (SiO2 / Fe3O4) in NCs 3-14 according to the UATR-FTIR data. A note “(Supplementary Materials, Figure S1)” has been added (line 291).

  • Paragraph 3.2.2. In this paragraph there is a general introduction on the difficulties of evaluating the data from fluorescence. The same ideas are further discussed in the text. Information is repeated.

Reply: The fragment in lines 397-400 has been removed.

  • Figure (6a) the nature of the peak at 530 nm is not discussed in the text. Can it influence the values of the 480 nm peak?

Reply: The shoulder at 530 nm is characteristic of the spectrum of doxorubicin, as evidenced by the presence of vibrational levels in the luminescence spectrum (Figure 5b in the revised manuscript); therefore, it does not affect the value of the peak at 480 nm.

On behalf of all coauthors.

Sincerely yours,

Dr. Alexander M. Demin

Reviewer 2 Report

The manuscript reports on the synthesis of an extensive list of nanocomposites comprising magnetite nanoparticles coated with silica, under the influence of different additives (PMIDA, APTMS, TEOS) plus a wide variety of preparation conditions, mainly aiming to evaluate the sorption and desorption of Doxorubicin (Dox). The data included in the submitted version of the manuscript are quite interesting and potentially can attract the attention of the Journal’s readership interested in exploring the described material platform for drug delivery. The manuscript presents a fair English, although there is room to improve it after a careful reading (awkward strings, punctuation, missing words, etc), which is highly recommended. Nevertheless, the manuscript presents gaps in regard to technical aspects and the corresponding literature review. Therefore, to the best of my understanding, the authors should prepare a revised version of the manuscript, either incorporating or rebutting (one by one) the comments listed below, while uploading the cover letter file.

1. My first observation goes to the characterization of the synthesized magnetite (Fe3O4), as the phase described in the manuscript. Usually, a fresh precipitate of magnetite oxidizes (even partially) to e.g. maghemite. Then, comes the very first question: do the authors confirm the magnetite phase in the magnetic precipitate? If so, please include this information with supporting data. Secondly, the text includes the information that the as-synthesized magnetite nanoparticles are of the order of 10 nm (page-6, line-239). Certainly, the authors recorded TEM micrographs of the magnetite nanoparticles. If so, please include the particle size distribution and the curve fitting of it. Additionally, the authors should inform the mean diameter and the diameter dispersion extracted from the fitting using a particle size distribution function (please, describe it). Moreover, while preparing the particle size histogram for curve fitting, please check the use of Sturge’s criteria. To be more specific, Sturges’ (1929) criteria should be applied, i.e. the number of vertical columns (C) in the histogram is defined by the number of recorded particles (N). According to Sturges’ criteria, the total number of Classes (C) scales with the number (N) of assessed particles via C=1+3.322*log(N). The authors can check the use of the Sturges’ relation in the paper by Aragon et al. in J. Phys. Cond. Matter vol. 27, paper 095301, 2015 (DOI: 10.1088/0953-8984/27/9/095301). The authors may consider the inclusion of this paper in the revised reference list. This point should be addressed in the revised version of the manuscript.

2. While reading the manuscript, the only practical reason to encapsulate magnetite nanoparticles in the described nanocomposites is the possible use of the end materials for contrast agents in MRI, which is related to the relaxometry data collected in Figure 5(c). Needless to mention a long list of applications of magnetic nanocomposites in the biomedical field. In this regard, it seems to be reasonable to provide extra magnetic data on a collection of as-prepared samples, such as the hysteresis cycles collected in Figure 3. However, the provided magnetic data (Figure 3) is not minimally discussed, even the change in the saturation magnetization of the samples due to the encapsulation process (obviously after removing the mass content of the diamagnetic material), thus emphasizing the impact of the coating material in the long-range magnetic ordering. Moreover, as the TEM pictures show (e.g. Figure 1(a)), magnetic particle-particle interaction within the silica-based template does influence the end magnetic behavior, including saturation magnetization, remanence and coercive field (check how much they change in the insets of Figure 3), and nearby proton relaxation times (T1 and T2). As in the present context, the magnetic data seems to be marginal in the authors’ understanding, it would be better to move Figure 3 to the Supplementary file. This is a way to avoid a comprehensive discussion.

3. On page-3, first paragraph of section 2.4., four samples (NCs 11-14) gave rise to three samples (NCs 15-17). Is there a missing sample? Is this a mistake?

4. On page-8, line-271, the authors wrote “…better hydrodynamic parameters”. What is the meaning of this statement? What are better and worse hydrodynamic parameters?

5. According to the authors the NC 18 sample presents a promising evaluation for use as a contrast agent in MRI (page-10, lines 374-376). However, according to the data collected in Figure 7, the NC 18 sample is positioned at the low end of Dox release (Figure 7(b)), although positioned at the high end of Dox sorption (Figure 7(a)). Actually, Figure 7 collects interesting data regarding sorption/desorption data. Unfortunately, a comparative discussion of the data is quite naïve in the present version of the manuscript, which is disappointing. Moreover, it is highly recommended the authors deepen the discussion of the data reported in Figure 7 in view of the expected application of the NCs for Dox delivery. For instance, a more detailed model picture (supported by a scheme) of the pH dependence of the Dox desorption would be extremely valuable and welcome. Therefore, to better assist the readership of the journal, the revised version of the manuscript should include the aforementioned discussion.

6. The authors dedicated a full section of the manuscript to the DFT calculations regarding the anchoring of the Dox molecule to different environments represented by the variety of NCs’ surfaces. Although extensive work has been performed in producing section 3.2.4. and binding energies (Eb) have been estimated, there is no discussion correlating the NUMBERS extracted from Eb calculation and the NUMBERS collected from the Dox sorption (data in Figure 7(a)). I’m confident the reader would be very much interested in a comprehensive discussion tightening these two pieces of information together. Therefore, to better assist the readership of the journal, the revised version of the manuscript should include this discussion.

7. While looking at the cell viability data provided in Figure 10 one wonders WHY the data were not analyzed in a more robust way, providing ground for correlating with the data collected in Figure 7(a) and (b). The very simple Hill model (proposed more than a century ago) is a very simple way to analyze cell viability to provide robust information regarding the toxicity of a nanomaterial, drug-delivering or not. Benchmark dose or DL50 can be easily extracted from the data collected in Figure 10. In this regard, it is highly recommended to the author carry out the proposed analysis and deepen their discussion of the cell viability connected to the sorption/desorption data. A recent analysis of the cell viability using a nanomaterial has been published by Li et al. in J. Mater. Chem. B, vol. 8, p. 2598, 2020 (DOI: 10.1039/c9tb02529d). The authors may consider the inclusion of this paper in the revised reference list. This point should be addressed in the revised version of the manuscript.

Indeed, I found the manuscript interesting but there are points to be revised, as indicated above, in terms of data discussion and reference list, that need to be bridged before this reviewer can support the manuscript published in the Pharmaceutics. The abstract and conclusions should be updated in the revised version of the manuscript.

Author Response

Comments and Suggestions for Authors:

The manuscript reports on the synthesis of an extensive list of nanocomposites comprising magnetite nanoparticles coated with silica, under the influence of different additives (PMIDA, APTMS, TEOS) plus a wide variety of preparation conditions, mainly aiming to evaluate the sorption and desorption of Doxorubicin (Dox). The data included in the submitted version of the manuscript are quite interesting and potentially can attract the attention of the Journal’s readership interested in exploring the described material platform for drug delivery. The manuscript presents a fair English, although there is room to improve it after a careful reading (awkward strings, punctuation, missing words, etc), which is highly recommended. Nevertheless, the manuscript presents gaps in regard to technical aspects and the corresponding literature review. Therefore, to the best of my understanding, the authors should prepare a revised version of the manuscript, either incorporating or rebutting (one by one) the comments listed below, while uploading the cover letter file.

  1. My first observation goes to the characterization of the synthesized magnetite (Fe3O4), as the phase described in the manuscript. Usually, a fresh precipitate of magnetite oxidizes (even partially) to e.g. maghemite. Then, comes the very first question: do the authors confirm the magnetite phase in the magnetic precipitate? If so, please include this information with supporting data. Secondly, the text includes the information that the as-synthesized magnetite nanoparticles are of the order of 10 nm (page-6, line-239). Certainly, the authors recorded TEM micrographs of the magnetite nanoparticles. If so, please include the particle size distribution and the curve fitting of it. Additionally, the authors should inform the mean diameter and the diameter dispersion extracted from the fitting using a particle size distribution function (please, describe it). Moreover, while preparing the particle size histogram for curve fitting, please check the use of Sturges’ criteria. To be more specific, Sturges’ (1929) criteria should be applied, i.e. the number of vertical columns (C) in the histogram is defined by the number of recorded particles (N). According to Sturges’ criteria, the total number of Classes (C) scales with the number (N) of assessed particles via C=1+3.322*log(N). The authors can check the use of the Sturges’ relation in the paper by Aragon et al. in J. Phys. Cond. Matter vol. 27, paper 095301, 2015 (DOI: 10.1088/0953-8984/27/9/095301). The authors may consider the inclusion of this paper in the revised reference list. This point should be addressed in the revised version of the manuscript.

Reply: Thank you very much for your appreciation of our work. To prove that the main phase of cores of the described NCs is magnetite and it is not changed during modification, we provided an electron diffraction pattern of the Dox-loaded nanocomposite (NC 18) (see Supplementary Materials, Figure S3a) as an example; we also provided a histogram of the size distribution for magnetic cores of this NC, which was obtained from the TEM data (see Supplementary Materials, Figure S3b).

  1. While reading the manuscript, the only practical reason to encapsulate magnetite nanoparticles in the described nanocomposites is the possible use of the end materials for contrast agents in MRI, which is related to the relaxometry data collected in Figure 5(c). Needless to mention a long list of applications of magnetic nanocomposites in the biomedical field. In this regard, it seems to be reasonable to provide extra magnetic data on a collection of as-prepared samples, such as the hysteresis cycles collected in Figure 3. However, the provided magnetic data (Figure 3) is not minimally discussed, even the change in the saturation magnetization of the samples due to the encapsulation process (obviously after removing the mass content of the diamagnetic material), thus emphasizing the impact of the coating material in the long-range magnetic ordering. Moreover, as the TEM pictures show (e.g. Figure 1(a)), magnetic particle-particle interaction within the silica-based template does influence the end magnetic behavior, including saturation magnetization, remanence and coercive field (check how much they change in the insets of Figure 3), and nearby proton relaxation times (T1 and T2). As in the present context, the magnetic data seems to be marginal in the authors’ understanding, it would be better to move Figure 3 to the Supplementary file. This is a way to avoid a comprehensive discussion.

Reply: According to your suggestion, in order to avoid a comprehensive discussion of the magnetic data, Figure 3 has been moved to the Supplementary Materials file (Figure S2).

  1. On page 3, first paragraph of section 2.4., four samples (NCs 11-14) gave rise to three samples (NCs 15-17). Is there a missing sample? Is this a mistake?

Reply: Those were misprints. The text has been revised; line 134: “A solution of Dox (10 mg) in water (1 mL) was added to a solution of NCs 3-8 and 9–14…”; line 137: “…to afford NCs 15–26”.

  1. On page-8, line-271, the authors wrote “…better hydrodynamic parameters”. What is the meaning of this statement? What are better and worse hydrodynamic parameters?

Reply: We have revised the text, lines 278-280: “In general, in aqueous solutions, the synthesized NCs obtained using PMIDA had smaller hydrodynamic sizes (Figure 1b), which is more preferable for their medical applications.”

  1. According to the authors the NC 18 sample presents a promising evaluation for use as a contrast agent in MRI (page-10, lines 374-376). However, according to the data collected in Figure 7, the NC 18 sample is positioned at the low end of Dox release (Figure 7(b)), although positioned at the high end of Dox sorption (Figure 7(a)). Actually, Figure 7 collects interesting data regarding sorption/desorption data. Unfortunately, a comparative discussion of the data is quite naïve in the present version of the manuscript, which is disappointing. Moreover, it is highly recommended the authors deepen the discussion of the data reported in Figure 7 in view of the expected application of the NCs for Dox delivery. For instance, a more detailed model picture (supported by a scheme) of the pH dependence of the Dox desorption would be extremely valuable and welcome. Therefore, to better assist the readership of the journal, the revised version of the manuscript should include the aforementioned discussion.

Reply: The relevant discussion concerning these issues has been added to the text of subsection 3.2.3. Quantitative Assessment of the Dox Content in NCs after Sorption / Desorption (lines 510-513 and 516-520).

  1. The authors dedicated a full section of the manuscript to the DFT calculations regarding the anchoring of the Dox molecule to different environments represented by the variety of NCs’ surfaces. Although extensive work has been performed in producing section 3.2.4. and binding energies (Eb) have been estimated, there is no discussion correlating the NUMBERS extracted from Eb calculation and the NUMBERS collected from the Dox sorption (data in Figure 7(a)). I’m confident the reader would be very much interested in a comprehensive discussion tightening these two pieces of information together. Therefore, to better assist the readership of the journal, the revised version of the manuscript should include this discussion.

Reply: It should be emphasized that due to the complexity of the objects, Dox Complexes with a Small SiO2 Cluster and Its Conjugates with APS and PMIDA, our results on the correspondence of Eb values to LC values are of a qualitative character. These results testify, first of all, to the correctness of our approach to calculations and allow us to make reasonable assumptions about the bonding nature in the studied complexes. A discussion of the correspondence between Eb values and LC values has been added to the text of section 3.2.4. DFT Calculations of the Total Electronic Energies of Dox Complexes with a Small SiO2 Cluster and Its Conjugates with APS and PMIDA (lines 558-568).

  1. While looking at the cell viability data provided in Figure 10 one wonders WHY the data were not analyzed in a more robust way, providing ground for correlating with the data collected in Figure 7(a) and (b). The very simple Hill model (proposed more than a century ago) is a very simple way to analyze cell viability to provide robust information regarding the toxicity of a nanomaterial, drug-delivering or not. Benchmark dose or DL50 can be easily extracted from the data collected in Figure 10. In this regard, it is highly recommended to the author carry out the proposed analysis and deepen their discussion of the cell viability connected to the sorption/desorption data. A recent analysis of the cell viability using a nanomaterial has been published by Li et al. in J. Mater. Chem. B, vol. 8, p. 2598, 2020 (DOI: 10.1039/c9tb02529d). The authors may consider the inclusion of this paper in the revised reference list. This point should be addressed in the revised version of the manuscript.

Reply: An appropriate discussion has been added to the text of subsection 3.3. Cytotoxicity Assessment (lines 578-581 and 587-592), including the IC50 values calculated for NCs 15-26 (Table 2, Figure S10). Ref. [97] proposed by the reviewer is cited in the manuscript.

Indeed, I found the manuscript interesting but there are points to be revised, as indicated above, in terms of data discussion and reference list, that need to be bridged before this reviewer can support the manuscript published in the Pharmaceutics. The abstract and conclusions should be updated in the revised version of the manuscript.

Reply: The abstract and conclusions have been updated.

On behalf of all coauthors.

Sincerely yours,

Dr. Alexander M. Demin

Round 2

Reviewer 2 Report

The revised version of the manuscript reporting on the synthesis of an extensive list of nanocomposites comprising magnetite nanoparticles coated with silica, under the influence of different additives (PMIDA, APTMS, TEOS) plus a wide variety of preparation conditions, satisfactorily included the points I raised in my previous report. Then, to the best of my understanding, the present version of the manuscript should be accepted for publication in Pharmaceutics.